# Clinical study of ultrasound-guided minimally invasive catheterization combined with compound cortex phellodendri fluid in the treatment of lactational breast abscess

**Na Wang** *[*], **Lili Gong** [*], **Chunmei Ye**

Department of Breast, Wuhan Children's Hospital (Wuhan Maternal and Child Healthcare Hospital), Tongji Medical College, Huazhong University of Science & Technology, Wuhan, China

[*] These authors contributed equally to this work.
* wangna@zgwhfe.com

## Abstract

### Objective

To retrospectively analyze the clinical practicability and value of ultrasound-guided minimally invasive catheterization combined with compound Phellodendron Phellodendri liquid in the treatment of breast abscess during lactation.

### Methods

139 patients with lactational breast abscess discharged from our hospital from January 2021 to November 2023 were selected. We divided them into groups according to treatment methods, analyzed whether there were statistical differences in observation indexes among groups and the risk factors affecting breastfeeding rate and treatment satisfaction.

### Results

We found that numerical rating scale(NRS) score and incidence of breast fistula in group A were significantly lower than other, the continuous decrease of postoperative drainage in group A was higher than other, there were significant differences among groups (p<0.001). Univariate analysis showed that recovery time, drainage tube placement time, postoperative redness and swelling regression time, scar length, and VAS score of six groups were statistically significant (p<0.001). We found that the overall satisfaction and the rate of continued breastfeeding in group A (96.2%) were higher than other, the differences were statistically significant(p<0.05). Logistic regression analysis revealed that the significant risk factors influencing treatment satisfaction included the time of drainage tube placement, postoperative redness and swelling regression time, treatment group, surgical method, NRS score on the first day after operation, postoperative drainage volume, healing time, scar length, flushing drugs, and VAS score. Postoperative redness and swelling regression time, treatment group, operation method and VAS score are all risk factors that influence the outcome of breastfeeding.

**Data Availability Statement:** All relevant data are within the manuscript and its Supporting Information files.

**Funding:** Hospital Level Research Project of Wuhan Children's Hospital (2021FE019).

**Competing interests:** The authors have declared that no competing interests exist.

## Conclusion

Ultrasound-guided minimally invasive catheterization combined with compound cortex phellodendri fluid in the treatment of breast abscess during lactation can not only reduce the pain caused by dressing change, but also offer numerous advantages, including shorter healing time, beautiful appearance, lower incidence of breast fistula, high satisfaction and high rate of continued breastfeeding.

## 1. Introduction

Breastfeeding is a scientifically endorsedparenting method promoted by the World Health Organization and the United Nations Children's Fund, that is of great significance to both parturients and infants. The advantages of breastfeeding are as follows: (1) ensuring the healthy growth of infants [1]; (2) reducing the incidence of breast cancer, obesity, diabetes, and cardiovascular disease, and so on [2]. Recommended breastfeeding for 2 years [3], aims to ensure the healthy growth of the baby. The causes of breastfeeding failure include breastfeeding mastitis and breast abscess, and the progress of breastfeeding mastitis can lead to breast abscess [4]. Patients with breast abscess during lactation have a high rate of stopping breastfeeding, which seriously affects the healthy growth of infants. Lactation abscess should be treated with appropriate antibiotics and ultrasound-guided aspiration of pus [5]. For patients with breast abscess during lactation, it is required to have a better aesthetic appearance of the breast while preserving the lactation function. However, the traditional catheter drainage still has some complications, including slow postoperative recovery, ahigh rate of breastfeeding cessation, scar formation, pain during dressing change, and so on. Although the puncture and irrigation method have achieved good results in the treatment of abscess, repeated punctures have brought physical pain and psychological fear to the patients. If the puncture fails, the patient still needs further surgical treatment. Therefore, it is urgent to find the best treatment for breast abscess in contemporary women. At the same time, it can not only ensure beauty and retain lactation function, but also achieve painless and other advantages.

At present, there are no report on the minimally invasive operation of breast abscess combined with Compound Cortex Phellodendri fluid in the treatment of breast abscess during lactation. Based on the existing medical records of breast abscess during lactation in our department, the differences in the treatment of various breast abscesses were retrospectively analyzed in order to find the best treatment for patients with breast abscess during lactation. It is expected that while retaining the function of breastfeeding and beauty, the pain during change of dressing will be reduced to a minimum. The discovery of this treatment will can benefit patients, infants and society.

## 2. Materials and method

### 2.1 General information

After obtaining ethical permission, we accessed our hospital's database on January 26, 2024. A total of 139 patients with breast abscess diagnosed in our hospital from January 1, 2021, to November 30, 2023, were collected. Inclusion criteria: (1) female patients; (2) patients who meet the diagnostic criteria for breast abscess during lactation; (3) minimally invasive catheterization of breast abscess or abscess incision catheterization at our hospital; (4) patients with newly diagnosed breast abscess and complete medical records. Exclusion criteria: (1)

complicated with other inflammatory or malignant tumors; (2) diseases that may affect inflammatory indexes, such as infectious diseases, hematological diseases, and autoimmune diseases, before operation; (3) patients with non-lactation breast abscess; (4) patients with blood coagulation dysfunction; (5) patients with hypertension, diabetes, and psychiatric diseases; (6) patients with breast cancer diagnosed by pathology;(7) puncture and irrigation of abscess. (8) those whose medical records are missing. This study was approved by the Ethics Committee of our hospital (Ethical examination and approval number: 2024R011-E01).

## 2.2 Research methods

**2.2.1 Basic information.** Within one year of the limited approval period of the ethical review, we can access information that can identify individual participants during or after data collection. Our study was retrospective, and the Ethics Committee approved it to be exempt from requiring informed consent. Consult the electronic medical record system of our hospital (outpatient+hospitalization system) to gather patients' clinically related medical records, including preoperative and postoperative medical records, as well as follow-up data. Preoperative medical records should encompass chief complaints, number of births, duration of breastfeeding, etc. Postoperative medical records should detail surgical methods, postoperative drainage changes, pain scores, and follow-up data should include satisfaction levels, breast feeding status, scar scores, and other relevant information.

**2.2.2 Main observation indicators.** The deadline for outpatient or telephone follow-up for all patients is January 20, 2024. Observation indicators: (1) General indicators: the days of hospitalization, recovery time, reoperation rate, patient satisfaction rate, and daily drainage were recorded and statistically analyzed. (2) Indexes of inflammation: the patients in the study were collected fasting peripheral venous blood before operation and in the morning of the third day after operation, and blood analysis was carried out by using Hisenmekang XN3000 automatic hematology analyzer and instrument supporting reagents in our hospital. The inflammatory indexes of the patients included in the study were calculated by excel table: WBC, neutrophils, hypersensitive C-reactive protein, calcitonin and pan-immune-inflammation value (PIV) = (neutrophils×platelets×monocytes) /lymphocytes; systemic immune-inflammation index (SII) = platelets×neutrophils/lymphocytes; systemic inflammation response index (SIRI) = neutrophils×monocytes / lymphocytes. The range of normal value refers to the standard of laboratory department of our hospital. (3) pain index: the pain score during daily dressing change was recorded by numerical rating scale(NRS) (Table 1). (4) Breastfeeding index: the occurrence and time of breast fistula in each group were recorded,

**Table 1. Numerical Rating Scale (NRS).**

| Project | score |
|---|---|
| analgesia | 0 |
| Slight pain, bearable | 1 |
| | 2 |
| | 3 |
| The pain affects sleep and is bearable | 4 |
| | 5 |
| | 6 |
| Intense pain, unbearable pain, affecting appetite and sleep | 7 |
| | 8 |
| | 9 |
| | 10 |

and the breast feeding rate of postoperative patients was recorded and statistically analyzed. (5) Local skin evaluation: a, for patients with local skin redness and swelling, record the time of disappearance of redness and swelling; b, the length of postoperative scar; c, whether the breast is deformed. (5) follow-up: after discharge from hospital, the patients were followed up for 6 months, and the follow-up time was the third day, the seventh day, 1 month, 3 months and 6 months after discharge. Patients who were not followed up at the outpatient clinic were followed up by telephone, including: the re-incidence of abscess; patient satisfaction rate; breast-feeding rate; whether there is scar, if there is scar, the scar was evaluated by the Vancouver Scar Scale (VAS) (Table 2). At the same time, during the follow-up, patients are informed that if a hard mass is found, please see a doctor in time, and we will give the patient breast dredging physiotherapy treatment in our hospital in time to avoid the recurrence of abscess.

**2.2.3 Group.** The patients included in the study were divided into groups based on the mode of operation and postoperative irrigating drugs:group A (minimally invasive catheterization +Compound Phellodendron Phellodendri liquid), group B (minimally invasive catheterization+Kangfuxin liquid) and group C (minimally invasive catheterization +normal saline), group D (abscess incision catheterization+Compound Phellodendron Phellodendri liquid), group E (abscess incision catheterization + Kangxin liquid) and group F(abscess incision catheterization + normal saline). All patients in 6 groups were treated with emergency operation on the day of admission or the second day after admission. We checked the blood routine before using antibiotics before operation, improved the relevant preoperative examination, and taken purulent secretions for bacterial culture. All patients in the six groups received sufficient antibiotics, intravenous anti-inflammatory treatment, and underwent surgery under general anesthesia.

1. Group A, B and C: vacuum-assisted minimally invasive surgery was performed, and the incision was selected to avoid the weakest skin, rupture, or edema on the surface of the abscess. According to aesthetic principles, the incision was made on the areola or submammary fold, and with a length of approximately3mm. Under the guidance of ultrasound, the minimally invasive needle was inserted into the abscess cavity to fully absorb the pus and

**Table 2. The Vancouver Scar Scale.**

| Project | index | score |
|---|---|---|
| Pigmentation (0–2) | Normal | 0 |
| | Hypopigmentation | 1 |
| | Hyperpigmentation | 2 |
| Vascularity (0–3) | Normal | 0 |
| | Pink | 1 |
| | Red | 2 |
| | Purple | 3 |
| Pliability (0–5) | Normal | 0 |
| | Supple | 1 |
| | Yielding | 2 |
| | Firm | 3 |
| | Banding | 4 |
| | Contracture | 5 |
| Height (0–3) | Normal (flat) | 0 |
| | 0–2 mm | 1 |
| | 2–5 mm | 2 |
| | >5 mm | 3 |

remove the septum and necrotic tissue of the abscess wall simultaneously. The flushing drug was injected into the abscess cavity and washed repeatedly, and the disposable negative pressure drainage ball (during the sterilization period) was placed and fixed in the pus cavity under the guidance of ultrasound.

2. Group D, E and F: traditional abscess incision technique was employed. The incision was carefully placed to avoid the weakest or ruptured skin on the surface of the abscess. The length of the incision was approximately 2cm. The procedure involved cutting through the skin and subcutaneous tissue sequentially, exploringthe abscess cavity, fully opening the septum, removing necrotic tissue, repeatedly injecting the flushing drug into the abscess cavity, and inserting a disposable negative pressure drainage ball (during the sterilization period) into the abscess cavity and securing it in place.

3. Patients in six groups were treated with irrigation starting from the first day after the operation. Group A and D were repeatedly rinsed with100 ml of Compound Cortex Phellodendri fluid (Chinese medicine Z10950097)(produced by Shandong Hanfang Pharmaceutical Co., Ltd.), and the washing solution was retained in the cavity for 1 hour. For patients with local skin rupture, a wet compress was applied to the affected area once a day using Compound Cortex Phellodendri fluid coating. Group B and E were rinsed repeatedly with 100ml of Kangfuxin liquid, and the washing solution was kept in the cavity for 1 hour. For patients with local skin rupture, a daily application of Kangfuxin with local skin wet compress was used. Group C and F were rinsed repeatedly with 100ml of normal saline. For individuals with local skin rupture, apply normal saline for wet compress on the affected area once a day. If there is an allergic reaction in the application of cortex phellodendri fluid or Fangfuxin liquid, and the symptoms are not relieved, the drug flushing treatment shall be suspended, and for those with severe allergic reaction, anti-allergic treatment will be used and rinse with normal saline. (the above drugs and consumables are within the period of validity and sterilization).

## 2.3 Statistical methods

The data of this retrospective study were statistically analyzed by SPSS27 statistical software. Shapiro-Wilk test method and histogram were used to check whether the measurement data conformed to the normal distribution. The measurement data in accordance with normal distribution are expressed as mean±standard deviation, and Ane-Way Analysis Of Variance (ANOVA) is used for comparison between groups. The measurement data that do not conform to the normal distribution are expressed by Median (M) and Inter-Quartile Range(IQR), and Kruskal-Wallis rank sum test is used for comparison between groups. The counting data were expressed as the number of cases (percentage) [n (%)]. Chi-square ($\chi$2) test or Fisher exact probability test was used for comparison between groups. Logistic regression model was used to analyze the independent risk factors of breastfeeding outcome and satisfaction. And use STATA 15 statistical software to draw forest map to analyze the risk factors in logistic regression analysis. The difference was statistically significant ($p < 0.05$).

## 3 Results

**3.1** A total of 139 patients, aged between 21 and 39 years old, were included in this retrospective study, the median age was 29 years old, and the median size of the purulent cavity was 5 (Q1 = 4, Q3 = 6). The preoperative data of each group were statistically analyzed. There was no statistical significance in the age among the six groups(F = 1.78, p = 0.121). According to the

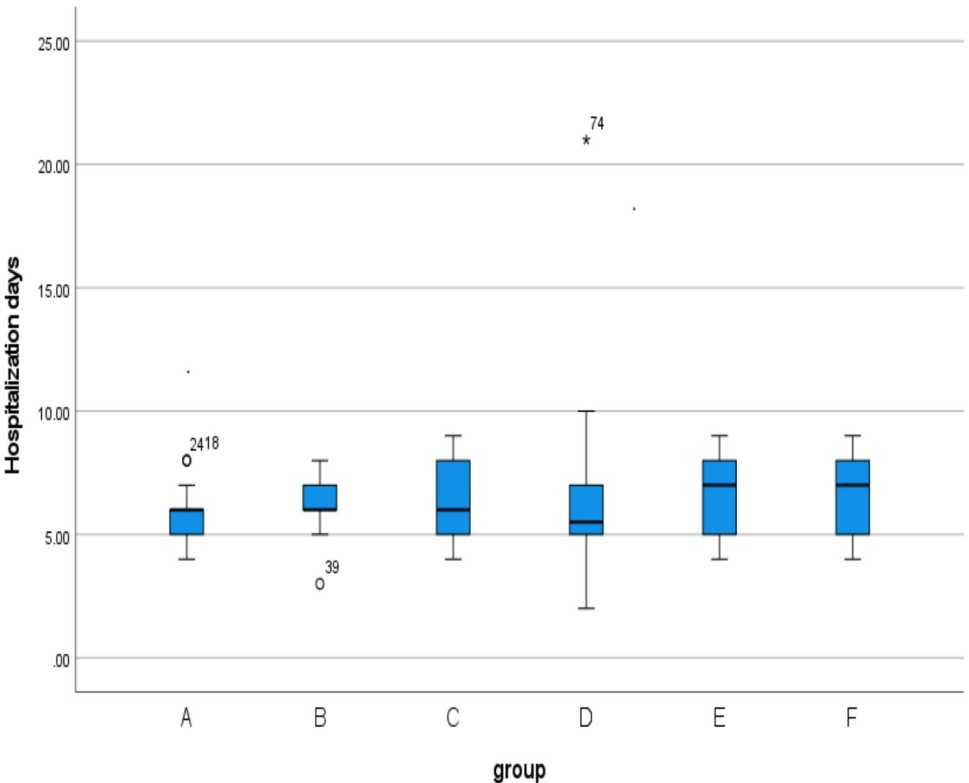

**Fig 1. Box plot show that there are no differences between different groups of the hospitalization days.**

box plot of non-normal distribution measurement data(Figs 1–3), there was no statistical significance in the time of onset (H = 5.666,p = 0.340), hospitalization days(H = 9.366, p = 0.095), and the size of pus cavity(H = 2.005, p = 0.848)in each group. The abnormal value of hospitalization days in group D was found in the box chart. Looking back at the medical records, it was found that the patient's actual hospital stay was 21 days, and the actual recovery time was 14 days. During the hospitalization, milk stasis occurred in the contralateral breast. Physiotherapy techniques such as breast massage and electrotherapy were implemented in our department, while also emphasizing the promotion of breastfeeding. The contralateral breast was not aggravated after treatment and the patient was cured and discharged from the hospital. There was no significant difference in lactation time, parity, and type of purulent cavity among the six groups (p > 0.05) (Table 3). In the observation of postoperative breast fistula, there was a significant difference in the occurrence of breast fistula among each group (p < 0.05) (Table 4). There were fewer breast fistulas in group A (7.7%).

**3.2** Kruskal-Wallis rank sum test was performed for the NRS scores of 139 patients before and after surgery (Table 5). There was no significant difference in preoperative NRS scores among the six groups (H = 3.014, p = 0.698). The NRS scores of the patients were evaluated during the dressing change for three days after the operation, and the NRS scores of the six groups showed statistically significant differences three days after the operation (p < 0.001). According to the box chart of postoperative NRS scores with statistical significance, when comparing the quartile and median, it was observed that the NRS score of group A was significantly lower than that of the other five groups (Figs 4–6).

**3.3** The Kruskal-Wallis rank sum test was performed on the postoperative drainage volume of 139 patients. Since some patients may have removed the drainage tube on the 3rd day after

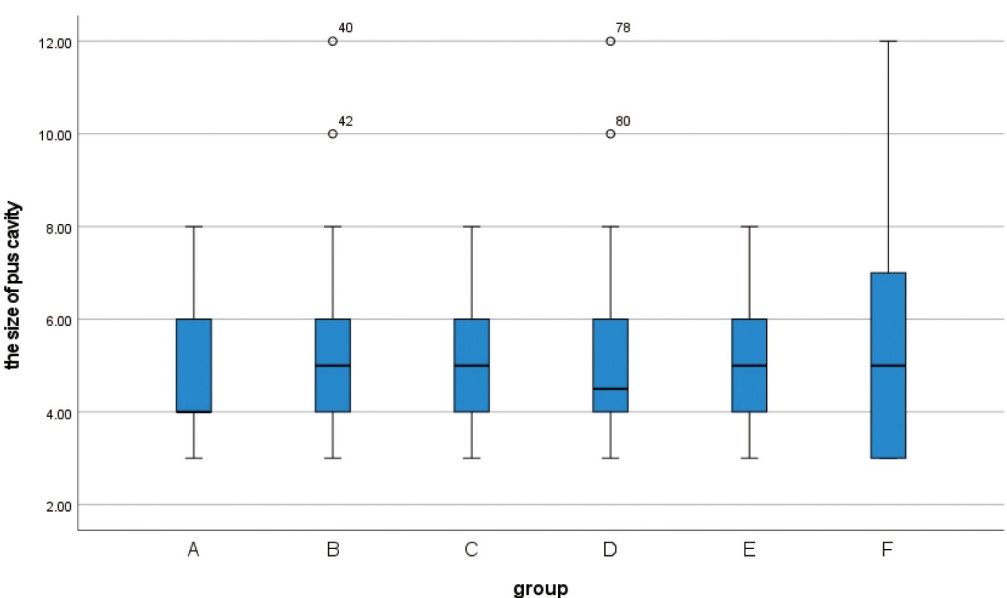

**Fig 2. Box plot show that there are no differences between different groups of the size of pus cavity.**

the operation, or have been discharged from the hospital, the data for part of the drainage volume cannot be obtained. Therefore, we collected the drainage volume within 3 days after the operation for statistical analysis. The drainage volume at 3 days after the operation was statistically significant in all 6 groups (p<0.05) (Table 6). At the same time, the box chart was created (Figs 7–9), and the quartiles and median were compared. The drainage volume of group A was lower than that of the other five groups within 3 days after the operation, and the continuous decrease in postoperative drainage of group A was higher than that of the other five groups. After the second day post-operation, significant difference between groups will emerge, with some abnormal values appearing.

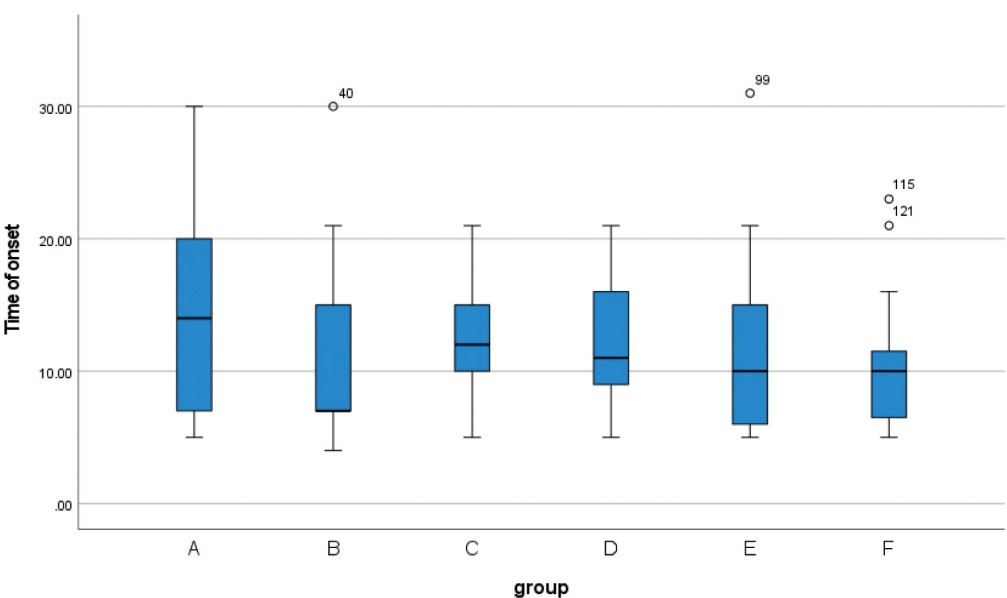

**Fig 3. Box plot show that there are no differences between different groups of the time of onset.**

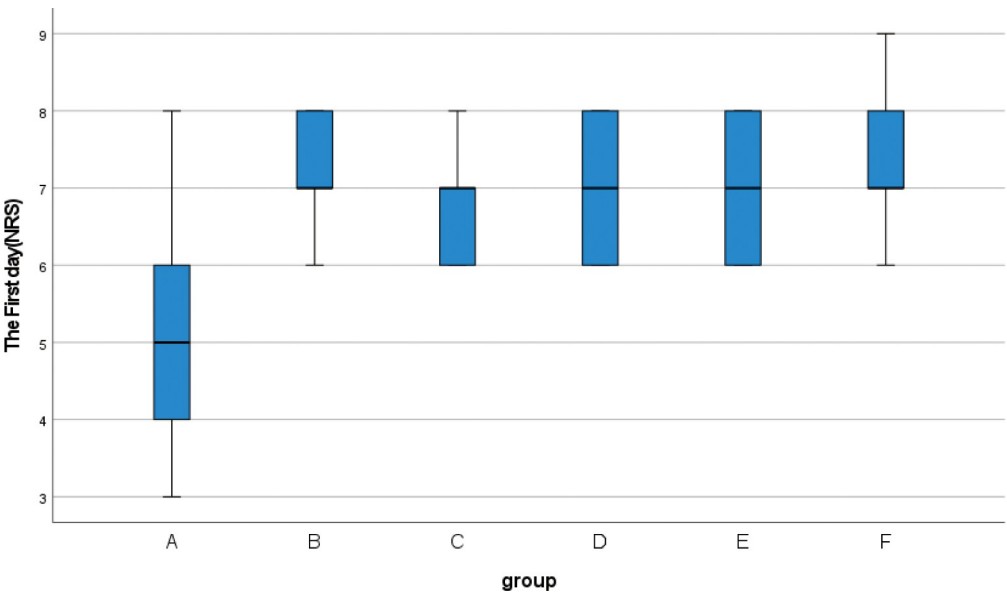

**Fig 4. Box plot show differences between different groups of NRS score on the first day.**

**3.4** 139 patients were followed up after the operation, and the measurement data in the follow-up data were analyzed using univariate analysis of variance (Table 7). Significant differences were found in recovery time, drainage tube placement time, postoperative redness and swelling regression time, scar length, and VAS score among the six groups (p <0.001). During the follow-up, we assessed the patients' satisfaction with this treatment, and a statistically significant difference was observed among the six groups (p <0.05). The overall satisfaction of patients in Group A was 96.2%, which was higher than that of theother groups. At the same time, the breastfeeding success rate of patients in group A (96.2%) was also significantly higher than that of theother groups (p <0.05).

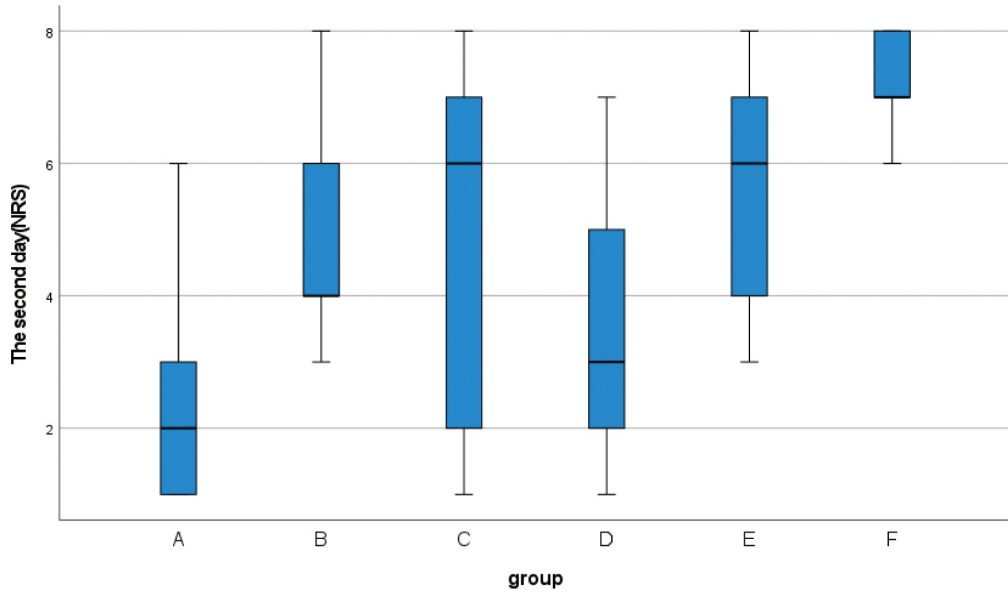

**Fig 5. Box plot show differences between different groups of NRS score on the second day.**

**Table 3. Baseline data of each group (mean±standard deviation)/n (%)/(M(IQR)).**

| Project/Group | minimal invasive (I) | | | Incision operation (II) | | | F/H/χ2 | p |
|---|---|---|---|---|---|---|---|---|
| | A | B | C | D | E | F | | |
| case | 26 | 21 | 21 | 22 | 22 | 27 | | |
| age(years) | 29.23±4.37 | 29.71±3.82 | 31.14±4.25 | 27.72±3.76 | 30.13±3.98 | 30.25±3.98 | 1.78 | 0.121 |
| Time of onset (day), M(IQR) | 14(13) | 7(8) | 12(6) | 11(8.5) | 10(9) | 10(6) | 5.67 | 0.340 |
| Hospitalization days (day), M(IQR) | 6(1) | 6(1.5) | 6(3) | 5.5(2) | 7(3) | 7(3) | 9.37 | 0.095 |
| Lactation time(day) | 39.50±14.07 | 48.52±20.31 | 42.19±14.57 | 54.23±33.90 | 39.27±12.39 | 45.26±14.27 | 2.02 | 0.080 |
| Involvement of central area of breast, n (%) | | | | | | | | |
| NO | 21 (80.8) | 16 (76.2) | 15 (71.4) | 14 (63.6) | 16 (72.7) | 18 (66.7) | 2.33 | 0.803 |
| YES | 5 (19.2) | 5 (23.8) | 6 (28.6) | 8 (36.4) | 6 (27.3) | 9 (33.3) | | |
| Purulent cavity size (cm), M(IQR) | 4(2.0) | 5(2.5) | 5(2.0) | 4.5(2.25) | 5(2.0) | 5(4.0) | 2.01 | 0.848 |
| Fetal number, n (%) | | | | | | | | |
| 1 | 20 (76.9) | 16 (76.2) | 15 (71.4) | 20 (90.9) | 16 (72.7) | 19 (70.4) | 3.55 | 0.615 |
| ≥2 | 6 (23.1) | 5 (23.8) | 6 (28.6) | 2 (9.1) | 6 (27.3) | 8 (29.6) | | |
| The septum of the purulent cavity, n (%) | | | | | | | | |
| NO | 17 (65.4) | 13 (61.9) | 13 (61.9) | 13 (59.1) | 15 (68.2) | 18 (66.7) | 0.58 | 0.989 |
| YES | 9 (34.6) | 8 (38.1) | 8 (38.1) | 9 (40.9) | 7 (31.8) | 9 (33.3) | | |

**Table 4. Mammary fistula (n(%)).**

| Project/Group | minimal invasive (I) | | | Incision operation (II) | | | χ2 | p |
|---|---|---|---|---|---|---|---|---|
| | A | B | C | D | E | F | | |
| case | 26 | 21 | 21 | 22 | 22 | 27 | | |
| mammary fistula | | | | | | | | |
| NO | 24 (92.3) | 17 (81.0) | 14 (66.7) | 12 (54.5) | 14 (63.6) | 17 (63.0) | 13 | 0.028 |
| YES | 2 (7.7) | 4 (19.0) | 7 (33.3) | 10 (45.5) | 8 (36.4) | 10 (37.0) | | |

**Table 5. Preoperative and postoperative NRS score (M (IQR)).**

| Project/Group | minimal invasive (I) | | | Incision operation (II) | | | H | p |
|---|---|---|---|---|---|---|---|---|
| | A | B | C | D | E | F | | |
| case | 26 | 21 | 21 | 22 | 22 | 27 | | |
| pre-operation | 7(2) | 7(2) | 7(2) | 7(2) | 7(2) | 7(1) | 3 | 0.698 |
| postoperation | | | | | | | | |
| The First day | 5(2) | 7(2) | 7(2) | 7(2) | 7(2) | 7(1) | 41 | <0.001 |
| The second day | 2(3) | 4(3) | 6(6) | 3(3) | 6(3) | 7(1) | 57 | <0.001 |
| The third day | 1(0) | 3(3) | 3(3) | 1(1) | 3(3) | 4(1) | 61 | <0.001 |

**Table 6. Postoperative drainage(ml) (M(IQR)).**

| Project/Group | minimal invasive (I) | | | Incision operation (II) | | | H | p |
|---|---|---|---|---|---|---|---|---|
| | A | B | C | D | E | F | | |
| case | 26 | 21 | 21 | 22 | 22 | 27 | | |
| The First day | 22(24.13) | 22(23.25) | 32(33.5) | 27(30) | 32.5(27) | 39(34) | 16 | 0.008 |
| The second day | 12(10.52) | 13(19.25) | 15(24) | 15.5(17.5) | 22(20) | 30(35) | 12 | 0.036 |
| The third day | 10(10.25) | 10(10) | 16(13.5) | 10(13.75) | 13.5(12.25) | 25(28) | 18 | 0.003 |

**Table 7. Relevant follow-up evaluation indicators after operation (mean±standard deviation)/n (%).**

| Project/Group | minimal invasive (I) | | | Incision operation (II) | | | F/χ2 | p |
|---|---|---|---|---|---|---|---|---|
| | A | B | C | D | E | F | | |
| case(n) | 26 | 21 | 21 | 22 | 22 | 27 | | |
| Recovery time(days) | 7.12±1.45 | 8.61±0.86 | 9.81±2.86 | 11.14±3.82 | 11.64±3.58 | 13.19±4.43 | 12.273 | <0.001 |
| Placement time of drainage tube(days) | 4.77±1.39 | 5.14±1.62 | 6.29±2.08 | 5.45±1.99 | 6.50±2.76 | 8.22±3.36 | 7.368 | <0.001 |
| Time of redness and swelling subsiding after operation(days) | 2.69±0.62 | 2.90±0.77 | 2.71±0.78 | 3.45±1.30 | 3.91±1.38 | 4.56±1.53 | 10.959 | <0.001 |
| Scar length(mm) | 2.5±0.51 | 2.43±0.51 | 2.52±0.51 | 14.41±1.01 | 14.36±0.95 | 14.30±1.07 | 1497 | <0.001 |
| VAS | 0.42±0.50 | 0.52±0.60 | 1.10±1.22 | 3.5±1.01 | 4.14±0.89 | 4.00±1.07 | 88.809 | <0.001 |
| Satisfaction, n (%) | | | | | | | | |
| Not satisfied | 1 (3.8) | 3 (14.3) | 6 (28.6) | 6 (27.3) | 8 (36.4) | 10 (37.0) | 11.441 | 0.043 |
| Satisfied | 25 (96.2) | 18 (85.7) | 15 (71.4) | 16 (72.7) | 14 (63.6) | 17 (63.0) | | |
| Lactation outcome, n (%) | | | | | | | | |
| Stop breastfeeding | 1 (3.8) | 1 (4.8) | 2 (9.5) | 5 (22.7) | 6 (27.3) | 8 (29.6) | 11.687 | 0.039 |
| breastfeeding | 25 (96.2) | 20 (95.2) | 19 (90.5) | 17 (77.3) | 16 (72.7) | 19 (70.4) | | |

## 3.5 Multi-factor analysis of satisfaction outcome and drawing of forest map

We take the satisfaction outcome after treatment as the dependent variable, assign the satisfaction outcome to 0 = dissatisfied, 1 = satisfied, and we utilize logistic regression analysis to analyze the statistically significant values of univariate analysis of variance in the study, the risk factors affecting the outcome of treatment satisfaction were analyzed. Through this retrospective study, we found that the relevant factors affecting treatment satisfaction included drainage tube placement time, postoperative redness and swelling regression time, treatment grouping, operation method, NRS score on the first day after the operation, postoperative drainage, healing time, scar length, irrigating drugs, and VAS score (Table 8). Using STATA15 software to analyze the risk factors affecting satisfaction outcomes in logistic regression analysis (Fig 10), we found that the placement time of postoperative drainage tube, postoperative redness and swelling regression

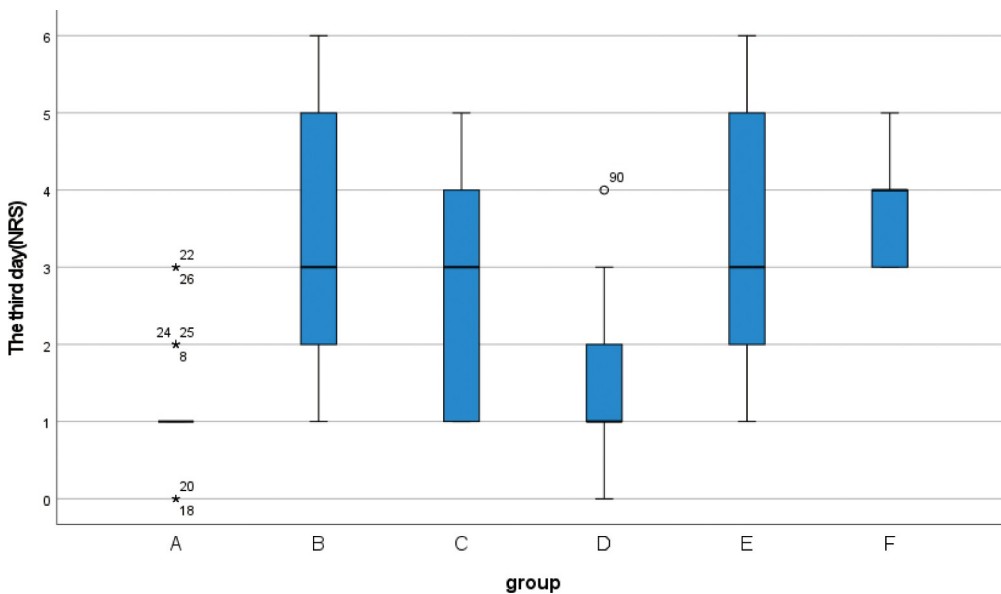

**Fig 6. Box plot show differences between different groups of NRS score on the third day.**

**Table 8. Logistic regression analysis affecting satisfaction outcome.**

| Project | Regression coefficient(B) | Wald | Exp(B)(95%CI) | p |
|---|---|---|---|---|
| Placement time of drainage tube | 0.163 | 5.007 | 0.849 (0.736–0.980) | 0.025 |
| Time of redness and swelling subsiding after operation | 0.279 | 3.840 | 0.757 (0.573–1.000) | 0.050 |
| Treatment grouping | 0.384 | 9.592 | 0.681 (0.534–0.868) | 0.002 |
| Mode of operation | 1.086 | 6.543 | 2.962 (1.289–6.806) | 0.011 |
| the NRS of postoperation | | | | |
| The First day | 0.477 | 5.363 | 0.620 (0.414–0.929) | 0.021 |
| The second day | 0.063 | 0.539 | 0.939 (0.793–1.111) | 0.463 |
| The third day | 0.225 | 3.285 | 0.799 (0.626–1.018) | 0.070 |
| postoperative drainage | | | | |
| The First day | 0.017 | 3.323 | 0.983 (0.965–1.001) | 0.068 |
| The second day | 0.022 | 6.132 | 0.978 (0.961–0.995) | 0.013 |
| The third day | 0.018 | 4.330 | 0.982 (0.966–0.999) | 0.037 |
| Recovery time | 0.014 | 5.197 | 0.901 (0.816–0.996) | 0.041 |
| Scar length | 0.089 | 6.391 | 0.915 (0.854–0.980) | 0.011 |
| Rinsing medication | 1.075 | 4.424 | 0.341 (0.125–0.929) | 0.035 |
| VAS | 0.319 | 7.996 | 0.727(0.582–0.907) | 0.005 |
| Lactation outcome | 0.035 | 0.005 | 0.966 (0.349–2.670) | 0.966 |

time, treatment group, NRS score on the first day after the operation, scar length, irrigating drugs, VAS score, and operation method had a significant influence on satisfaction. Among them, the influence of irrigation drugs and surgical methods was significant. The odds ratio (OR) value for irrigation drugs affecting satisfaction was 0.341, 95% CI 0.13–0.93. The odds ratio value for satisfaction, influenced by the mode of operation, was 2.96, 95% CI was 0.29–6.81.

## 3.6 Analysis of multiple factors affecting lactation outcome and drawing of forest map

We take the breastfeeding outcome after treatment as a dependent variable, assign a value of 0 = stopping breastfeeding, 1 = continuing breastfeeding, and use logistic regression analysis

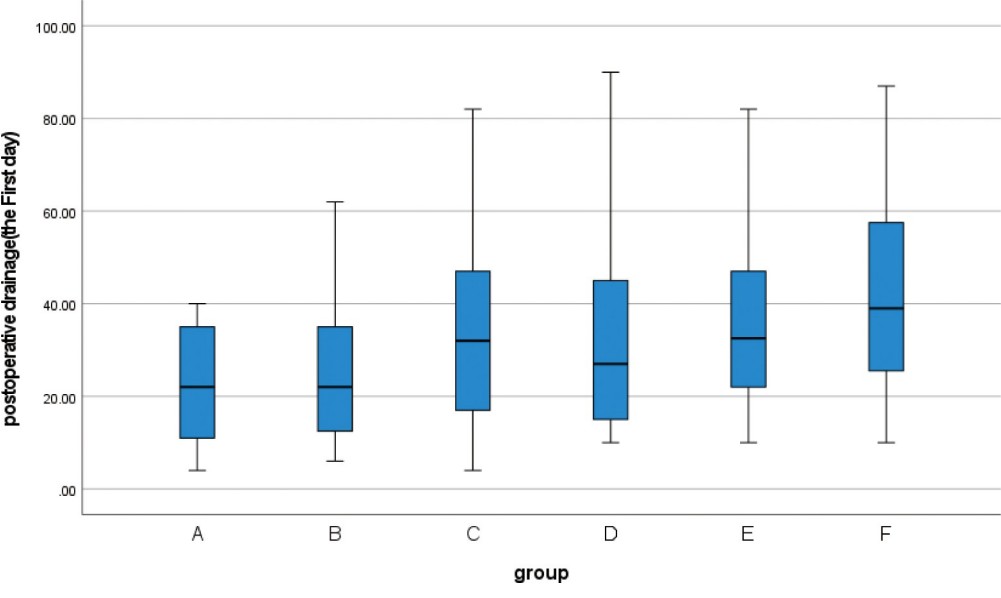

**Fig 7. Box plot show differences between different groups of postoperative drainage on the first day.**

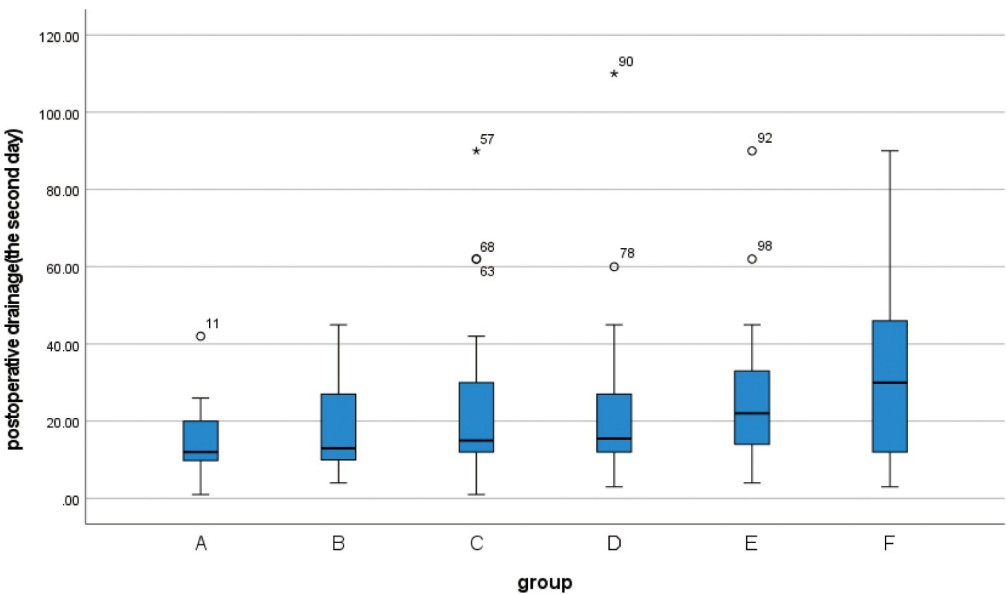

**Fig 8. Box plot show differences between different groups of postoperative drainage on the second day.**

to analyze the risk factors influencing breastfeeding outcomes. Through this study, it was found that the factors affecting the outcome of breastfeeding included the regression time of postoperative redness and swelling, the grouping of treatment, the mode of operation, the length of scar, and VAS score (Table 9). Using STATA15 software to draw forest map to analyze the risk factors affecting lactation outcome in logistic regression analysis (Fig 11), we found that postoperative redness and swelling regression time, treatment grouping, operation method, and VAS score had great influence on lactation outcomes.

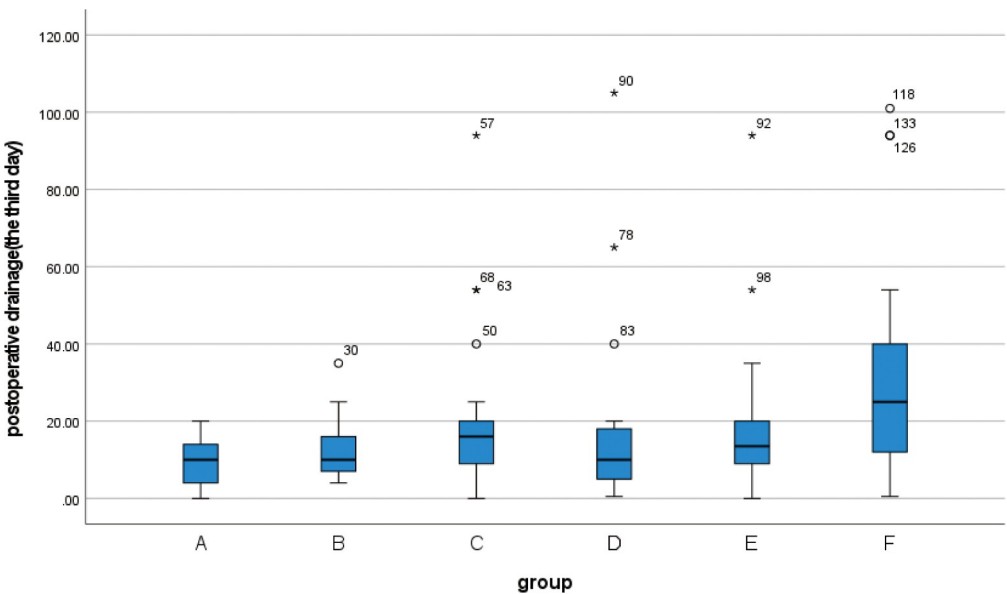

**Fig 9. Box plot show differences between different groups of postoperative drainage on the third day.**

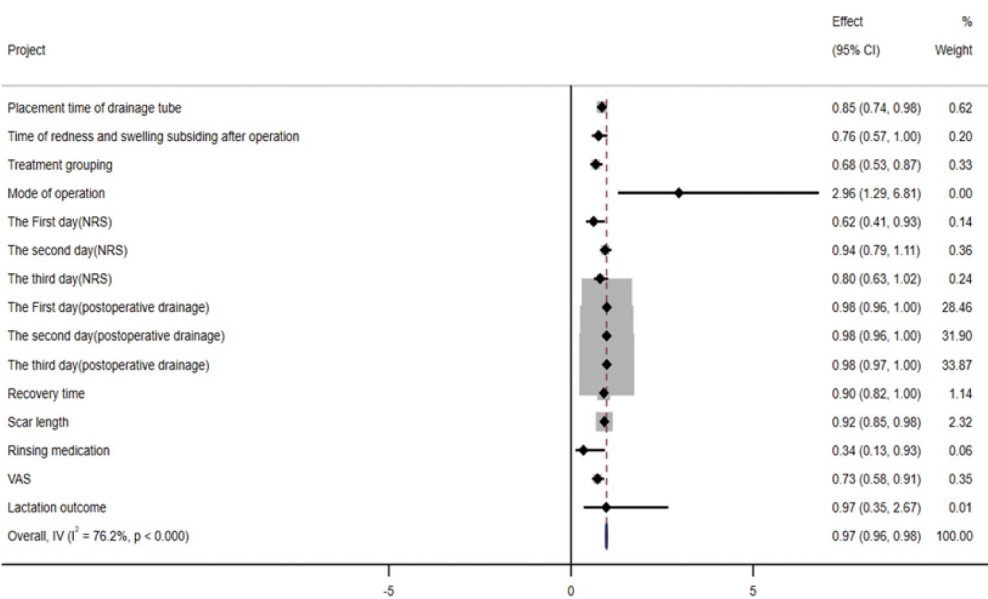

**Fig 10. Forest map of related factors affecting satisfaction.**

## 3.7 Inflammation index

Univariate analysis of variance showed that there was no significant difference in inflammatory indexes among the three groups (p> 0.05) (Table 10).

## 4 Discussion

Breast abscess during lactation is one of the primary reasons for breastfeeding interruption [6]. In the treatment of breast abscess, various breast specialists have differing opinions and strive

**Table 9. Logistic regression analysis affecting lactation outcome.**

| Project | Regression coefficient(B) | Wald | Exp(B)(95%CI) | p |
|---|---|---|---|---|
| Placement time of drainage tube | 0.073 | 0.811 | 0.930 (0.793–1.090) | 0.368 |
| Time of redness and swelling subsiding after operation | 0.415 | 7.141 | 0.660 (0.487–0.895) | 0.008 |
| Treatment grouping | 0.449 | 9.152 | 0.638 (0.477–0.854) | 0.002 |
| Mode of operation | 1.527 | 8.103 | 0.217 (0.076–0.622) | 0.004 |
| the NRS of postoperation | | | | |
| The First day | 0.359 | 2.603 | 0.698 (0.452–1.080) | 0.107 |
| The second day | 0.139 | 1.873 | 0.870 (0.713–1.062) | 0.171 |
| The third day | 0.156 | 1.264 | 0.855 (0.651–1.123) | 0.261 |
| postoperative drainage | | | | |
| The First day | 0.018 | 2.995 | 0.982 (0.962–1.002) | 0.084 |
| The second day | 0.012 | 1.558 | 0.988 (0.970–1.007) | 0.212 |
| The third day | 0.015 | 2.600 | 0.985 (0.968–1.003) | 0.107 |
| Recovery time | 0.1 | 3.207 | 0.905 (0.811–1.009) | 0.073 |
| Scar length | 0.013 | 8.527 | 0.878 (0.804–0.958) | 0.003 |
| Rinsing medication | 0.308 | 0.263 | 0.735 (0.226–2.386) | 0.608 |
| VAS | 0.396 | 8.607 | 0.673 (0.517–0.877) | 0.003 |
| satisfaction | 0.035 | 0.005 | 0.966 (0.349–2.670) | 0.946 |

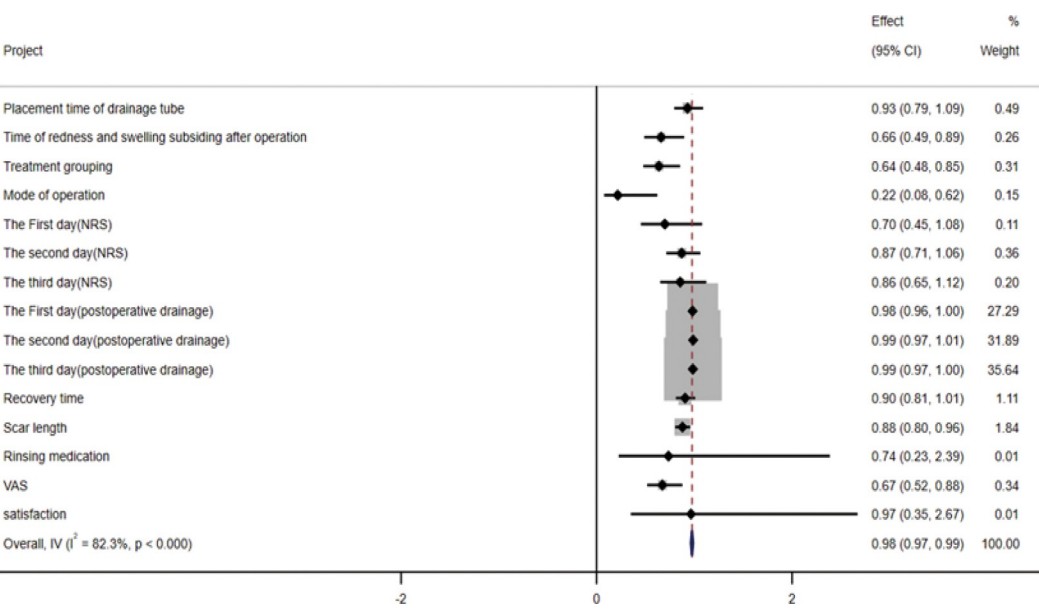

**Fig 11. Forest maps affecting lactation outcomes.**

to ensure a higher breastfeeding rate while treatment. Lam [7] believes that surgical treatment is the first choice for multilocular abscess with a long course of disease, purulent cavities larger than 5cm, and septum. Ulitzsch [8] retrospectively evaluated the treatment of breast abscess during lactation with the assistance of ultrasound guidance. They found that patients with abscess diameter less than 3cm can be treated by puncture, and patients larger than 3cm can be treated by catheterization, but it does not show that 3cm is the dividing point between the two schemes. Eryilmaz [9] thinks that patients with breast abscess diameter smaller than 5cm are suitable for puncture and irrigation treatment.

**Table 10. Related inflammation evaluation indexes before and after operation (mean±standard deviation).**

| Project/Group | minimal invasive (I) | | | Incision operation (II) | | | F | p |
|---|---|---|---|---|---|---|---|---|
| | A | B | C | D | E | F | | |
| case | 26 | 21 | 21 | 22 | 22 | 27 | | |
| pre-operation WBC | 11.603±3.982 | 9.635±2.973 | 9.697±2.978 | 10.711±3.371 | 10.282±2.841 | 11.007±3.461 | 1.257 | 0.286 |
| pre-operation PCT | 0.211±0.471 | 0.058±0.102 | 0.0561±0.103 | 0.063±0.099 | 0.059±0.100 | 0.201±0.453 | 1.609 | 0.162 |
| pre-operation CRP | 22.996±22.790 | 12.209±9.111 | 12.271±11.057 | 16.018±11.305 | 14.609±11.398 | 18.74±15.528 | 1.922 | 0.95 |
| pre-operation Neutrophils | 8.788±3.726 | 6.939±3.007 | 6.998±2.906 | 7.911±3.231 | 7.513±2.792 | 8.141±3.203 | 1.178 | 0.324 |
| pre-operation PIV | 1279.124±1056.872 | 835.236±730.087 | 822.475±726.753 | 1044.675±877.114 | 939.309±698.262 | 1074.358±930.118 | 0.952 | 0.45 |
| pre-operation SII | 1727.563±658.161 | 1418.384±792.779 | 1385.368±729.984 | 1605.124±771.904 | 1550.078±773.921 | 1644.792±577.273 | 0.796 | 0.555 |
| pre-operation SIRI | 3.525±2.469 | 2.662±2.072 | 2.473±1.879 | 3.087±2.187 | 2.966±1.943 | 2.815±1.968 | 0.723 | 0.608 |
| postoperation WBC | 5.961±0.911 | 6.571±1.529 | 6.132±1.597 | 6.156±1.507 | 6.411±1.559 | 6.631±1.622 | 0.82 | 0.538 |
| postoperation PCT | 0.038±0.015 | 0.041±0.014 | 0.034±0.018 | 0.038±0.017 | 0.038±0.017 | 0.040±0.016 | 0.464 | 0.803 |
| postoperation CRP | 9.987±5.782 | 7.000±5.515 | 7.950±5.744 | 7.194±5.413 | 6.977±5.153 | 5.955±4.315 | 1.717 | 0.135 |
| postoperation Neutrophils | 3.641±0.869 | 3.858±1.266 | 3.36±1.223 | 3.535±1.144 | 3.549±1.220 | 3.774±1.131 | 0.542 | 0.744 |
| postoperation PIV | 276.146±117.202 | 285.083±194.917 | 249.709±200.996 | 256.951±173.346 | 278.678±219.064 | 300.317±222.787 | 0.222 | 0.953 |
| postoperation SII | 818.991±247.446 | 729.968±314.337 | 637.54±318.505 | 693.689±343.537 | 678.18±308.265 | 645.119±236.846 | 1.301 | 0.267 |
| postoperation SIRI | 0.767±0.301 | 0.919±0.643 | 0.828±0.664 | 0.814±0.517 | 0.922±0.747 | 1.042±0.782 | 0.639 | 0.67 |

At present, minimally invasive techniques have been widely used in clinics, which can achieve the best curative effects with minimal trauma and invasion. They not only reduce the pain of patients, but also promote postoperative recovery and result in an aesthetic effect after the operation. Its advantages are more obvious in the treatment of breast diseases. Take advantage of minimally invasive advantages and apply them to the treatment of breast abscess [10]. The vacuum-assisted rotary cutting system is used to clean the pus and remove the necrotic tissue of the abscess wall simultaneously, which has many advantages, such as minimal trauma, quick healing, and short duration of the disease. The advantage of ensuring breastfeeding is beneficial to the growth and development of infants and young children. So ithas good clinical application value, but minimally invasive treatment of breast abscess has not been widely used in clinical practice. Currently, there are no reports on the application of compound cortex phellodendri fluid in the treatment of breast abscess. The ingredients of Compound Cortex Phellodendri fluid include Forsythia suspensa, Cortex Phellodendri, Flos Lonicerae, Taraxacum, and Scolopendra. It has a variety of functions, the most important of which include: (1) removing saprophytic muscle. (2) Anti-inflammation, detumescence, and analgesia effects: Cortex Phellodendri [11] and Scolopendra [12] have anti-inflammatory, detumescence, and analgesic effects. (3) The effects of heat-clearing and detoxification, sterilization, itching, and anti-exudation: Dandelion [12] and Forsythia suspensa [13, 14] contain sterols, which have antibacterial and bacteriostatic effects. (4) Improve non-specific immunity: Honeysuckle [15] can enhance monocyte-macrophage phagocytosis, antibacterial activity, and non-specific immunity. Forsythia and honeysuckle are commonly used medicinal combinations that can help clear away internal heat, detoxify sores, and promote skin health. Compound Cortex Phellodendri fluid has been used in the treatment of infectious diseases [11, 16] with remarkable efficacy and a high level of safety.

In clinical work, we typically utilize compound Cortex Phellodendri fluid, Kangfuxin liquid, or normal saline for postoperative irrigation in the treatment of breast abscess. This is done to compare the efficacy of various treatment approaches for breast abscess during lactation and determine the optimal treatment regimen. This retrospective study was designed. The baseline values of 139 patients, including the time of lactation, the number of births, the type of pus cavity, the time of onset, the days of hospitalization, and the size of the pus cavity, showed no significant difference (p >0.05). According to different treatment schemes, they are divided into six groups. It was found that there were fewer breast fistulas in group A (7.7%). At the same time, the NRS scores of the six groups were statistically analyzed, and we found that there was no significant difference in NRS scores among the six groups before the operation. We recorded the NRS scores during the dressing change for three consecutive days post-operation. We observed a statistically significant difference in the NRS scores of six groups three days after the operation (p <0.001). We also found that the NRS score of group A was significantly lower than the scores other five groups. Three days after the operation, the M (IQR) of group A was 5 (2), 2 (3), and 1 (0), respectively. Thus, it can be seen that Compound Cortex Phellodendri fluid can relieve pain in the treatment of breast abscess during lactation. During surgical treatment, we routinely insert a drainage tube into the abscess cavity of patients with breast abscess. Some patients had less drainage three days after the operation, may had removed the drainage tube, or had been discharged from the hospital, and in some cases, the drainage data could not be obtained. So, we collected the drainage volume within 3 days after the operation for statistical analysis. It was found that the drainage volume in group A within 3 days after the operation was lower than other five groups (p <0.05). The continuous decrease in postoperative drainage in group A was higher than in the other five groups. We also found that the total drainage volume in minimally invasive surgery for breast abscess was lower than that in breast abscess patients who underwent incision. This also reflects the effectiveness of minimally

invasive surgery and drainage in the treatment of breast abscess, which is consistent with the research findings of Fahrni [17] and other scholars.

After the patient was discharged from the hospital, we conducted regular follow-ups, including telephone and outpatient follow-up appointments. We chose 3 days, 1 week, 1 month, 3 months, and half a year after discharge. During the follow-up, we mainly observed the recovery time, drainage tube placement time, postoperative redness and swelling regression time, scar length, VAS score, breast fistula occurrence, satisfaction survey, and other relevant factors. We found that the follow-up indexes of Group A were better than those of the other five groups. We also found that the follow-up outcomes of the minimally invasive catheter drainage group were better than those of patients who underwent incision and catheterization. During the follow-up, we conducted a satisfaction survey. The overall satisfaction of patients in group A was 96.2%, which was higher than that of the other groups (p <0.05). We also analyzed the risk factors affecting the satisfaction in this retrospective study. We found that the placement time of the postoperative drainage tube, the time for redness and swelling regression after the operation, the treatment group, NRS score on the first day after the operation, scar length, irrigation drug, and VAS score, and operation method significantly influenced satisfaction. Among these factors, the influence of irrigation drug and operation method was particularly notable. The OR value of irrigation drug on satisfaction was 0.341. The 95%CI was 0.13–0.93, and the OR value of operation mode affecting satisfaction was 2.96, 95% CI 0.29–6.81. We observed the breastfeeding outcomes of patients and found that the rate of continued breastfeeding in group A (96.2%) was also higher than that in the other groups (p < 0.05). Similarly, we analyzed the risk factors affecting lactation outcomes in this retrospective study, including postoperative redness and swelling regression time, treatment grouping, operation method, scar length, and VAS score. Among these factors, the time for postoperative redness and swelling to regress, treatment grouping, surgical method, and VAS score significantly influenced the outcome of lactation. Scar length and VAS score may be related to hand style, so they cannot be completely used as risk factors for lactation outcome. The OR for the impact of postoperative redness and swelling regression time on lactation outcome is 0.66(95% CI 0.487–0.895). The OR value of the treatment group affecting the breastfeeding outcome was 0.638(95% CI 0.477–0.854), and the OR of the operation method affecting the breastfeeding outcome was 0.217(95% CI 0.076–0.622). In the course of treatment, we routinely administered an adequate amount of antibiotics. Univariate analysis of variance indicated that there was no significant difference in inflammatory indexes among the three groups (p > 0.05).

## 5 Conclusions

Through this retrospective study, we find that ultrasound-guided minimally invasive catheterization combined with compound cortex phellodendri fluid has a variety of advantages in the treatment of breast abscess during lactation, including: (1) short postoperative recovery time; (2) short postoperative drainage tube placement time; (3) postoperative beauty; (4)relieve pain, or no pain; (5) high satisfaction; (6) low incidence of breast fistula, high rate of continuous breastfeeding and low rate of stopping breastfeeding. At the same time, the time of disappearance of redness and swelling, the mode of treatment and the mode of operation are the common risk factors affecting the outcome and satisfaction of breastfeeding. Thus, it can be seen that this treatment of breast abscess can be safely applied in clinic. It can reduce the pain during dressing change to a minimum while retaining the functions of breastfeeding and beauty. The discovery of this treatment will benefit patients, infants and society. However, there are also deficiencies in this study, and the reasons for dissatisfaction of some patients include the

high cost of treatment. This retrospective study is a single-center study with a small sample size. In the later stage, we can increase the sample size and multicenter study to verify the advantages of ultrasound-guided minimally invasive catheterization combined with compound Phellodendron Phellodendri liquid in the treatment of breast abscess during lactation.

## Supporting information

**S1 Dataset.**
(XLSX)

## Acknowledgments

The author thanks all the patients and their families who were included in this study, as well as the colleagues from the medical records department and scientific research department of our hospital for their help in this study.

## Author Contributions

**Conceptualization:** Na Wang, Lili Gong, Chunmei Ye.

**Data curation:** Na Wang, Lili Gong.

**Formal analysis:** Na Wang.

**Investigation:** Na Wang.

**Methodology:** Na Wang, Lili Gong, Chunmei Ye.

**Project administration:** Na Wang, Lili Gong.

**Supervision:** Na Wang, Lili Gong, Chunmei Ye.

**Visualization:** Na Wang, Lili Gong, Chunmei Ye.

**Writing – original draft:** Na Wang, Lili Gong.

**Writing – review & editing:** Na Wang, Lili Gong, Chunmei Ye.

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
