## [Decision Letter · Decision Letter 0]

17 May 2024

PONE-D-24-05063Clinical Study Of Ultrasound-Guided Minimally Invasive Catheterization Combined with Compound Cortex Phellodendri Fluid in the Treatment of Lactational Breast AbscessPLOS ONE

Dear Dr. wang,

Thank you for submitting your manuscript to PLOS ONE. After careful consideration, we feel that it has merit but does not fully meet PLOS ONE’s publication criteria as it currently stands. Therefore, we invite you to submit a revised version of the manuscript that addresses the points raised during the review process.

We look forward to receiving your revised manuscript.

Kind regards,

Zhaoqing Du, Ph.D

Academic Editor

PLOS ONE

Journal Requirements:

Reviewers' comments:

Reviewer's Responses to Questions

**Comments to the Author**

1. Is the manuscript technically sound, and do the data support the conclusions?

Reviewer #1: Yes

Reviewer #2: Yes

Reviewer #3: Yes

2. Has the statistical analysis been performed appropriately and rigorously? 

Reviewer #1: Yes

Reviewer #2: Yes

Reviewer #3: Yes

3. Have the authors made all data underlying the findings in their manuscript fully available?

Reviewer #1: Yes

Reviewer #2: Yes

Reviewer #3: Yes

4. Is the manuscript presented in an intelligible fashion and written in standard English?

Reviewer #1: No

Reviewer #2: Yes

Reviewer #3: Yes

5. Review Comments to the Author

Reviewer #1: Plos One. manuscript TITLE: “Clinical Study of Ultrasound-Guided Minimally Invasive Catheterization Combined with Compound Cortex Phellodendri Fluid in the Treatment of Lactational Breast Abscess” presented some original but preliminary findings. The manuscript strengths ultrasound-guided minimally invasive catheterization combined with compound cortex phellodendri fluid in the treatment of breast abscess during lactation can not only reduce the pain caused by dressing change,at the same time, it has many advantages, such as short healing time, beautiful appearance, low incidence of breast fistula, high satisfaction and high rate of continued breastfeeding. However, the manuscript need to fix writing, better define cases, and be clearer in conclusions regarding findings and their implications.

Reviewer #2: The author creatively combined the minimally invasive operation with compound cortex phellodendri fluid to treat breast abscess during lactation. Compared to traditional treatment methods, minimally invasive operation with compound cortex phellodendri fluid obtained better treatment outcomes and higher satisfaction. While, I would like to know whether the treatment of breast abscess is chosen by the patient or by the doctor? If doctors decided the treatment method, what are the conditions under which they choose the treatment plan? By the way, I suggest the author to change all tables to standard three line tables.

Reviewer #3: Although this content is very suggestive, abstract lacks enough explanation. It is necessary for contents to come in abstract alone, but the minimum explanation is insufficient.

Six groups of treatment methods should be explained in abstract.

Thus, this paper is acceptable with minor revision.

6. PLOS authors have the option to publish the peer review history of their article (what does this mean?). If published, this will include your full peer review and any attached files.

Reviewer #1: No

Reviewer #2: **Yes: **He Tao

Reviewer #3: No

---

## [Author Response · Author response to Decision Letter 0]

30 May 2024

Dear Editors and Reviewers: 

Thank you very much for your comments and professional advice. Those comments are all valuable and very helpful for revising and improving our paper, as well as the important guiding significance to our researches. We have studied comments carefully and have made correction which we hope meet with approval. Meanwhile, the manuscript had be reviewed improve academic rigor of our article. We hope that our work can be improved again. Revised portion are marked in blue in the paper. The main corrections in the paper and the responds to the reviewer’s comments are as flowing:

Responds to the reviewer’s comments:

Reviewer #1: Plos One. manuscript TITLE: “Clinical Study of Ultrasound-Guided Minimally Invasive Catheterization Combined with Compound Cortex Phellodendri Fluid in the Treatment of Lactational Breast Abscess” presented some original but preliminary findings. The manuscript strengths ultrasound-guided minimally invasive catheterization combined with compound cortex phellodendri fluid in the treatment of breast abscess during lactation can not only reduce the pain caused by dressing change, at the same time, it has many advantages, such as short healing time, beautiful appearance, low incidence of breast fistula, high satisfaction and high rate of continued breastfeeding. However, the manuscript need to fix writing, better define cases, and be clearer in conclusions regarding findings and their implications.

The author’s answer: First of all, thank you for taking the time to review our manuscript, and special thanks to you for your approval. Here below we address the questions and suggestions raised by reviewer #1.

We revise our manuscript carefully and repeatedly to ensure that the manuscript has correct writing standards. At the same time, We re-examined our manuscript to make the grouping in the manuscript clearer. We carefully examined the conclusions in the manuscript and carefully revised the conclusions to make sure our findings were clearer. 

Thank you again for your comments. Once again, we look forward to your approval for our revision.

Reviewer #2: The author creatively combined the minimally invasive operation with compound cortex phellodendri fluid to treat breast abscess during lactation. Compared to traditional treatment methods, minimally invasive operation with compound cortex phellodendri fluid obtained better treatment outcomes and higher satisfaction. While, I would like to know whether the treatment of breast abscess is chosen by the patient or by the doctor? If doctors decided the treatment method, what are the conditions under which they choose the treatment plan? By the way, I suggest the author to change all tables to standard three line tables.

The author’s answer: thank you for taking the time to review our manuscript, and special thanks to you for your good comments. Here below we address the questions and suggestions raised by reviewer #2.

(1)Our study is a retrospective study. After the patients were admitted to the hospital, the doctor would inform them of their condition in detail, and evaluate the actual situation of the abscess of the breast by color Doppler ultrasound again, and inform the patients of the advantages and disadvantages of the two surgical treatment options. It is up to the patient to choose the specific operation plan. In the choice of flushing drugs, the patients are also informed of the advantages of the three flushing drugs and the unpredictable uncertainty. The patient chooses the flushing drugs. If an allergic reaction occurs in the course of treatment, stop the treatment immediately. At the same time, anti-allergic treatment needs to be done immediately.

(2)Thank you very much for pointing out the table problem in our manuscript. We have carefully examined our table that in our manuscript and changed all tables to standard three line tables at the same time.

Reviewer #3: Although this content is very suggestive, abstract lacks enough explanation. It is necessary for contents to come in abstract alone, but the minimum explanation is insufficient.

Six groups of treatment methods should be explained in abstract.

Thus, this paper is acceptable with minor revision.

The author’s answer: we are very grateful to reviewer #3 for his/her effort in reviewing our paper and his/her positive feedback. Here below we address the questions and suggestions raised by reviewer #3.

(1)We have re-written the abstract according to the Reviewer suggestion, so that it has enough explanation. As the journal has strict requirements on the number of words of the abstract, we use the limited number of words to maximize the introduction of our findings in the abstract. Hope to get your approval.

(2)Due to the limited number of words in the abstract, six groups of treatments are explained in detail in the text of the manuscript. Line141-148(p8) 

We tried our best to improve the manuscript and made some changes in the manuscript. These changes will not influence the content and frame work of the paper. 

We appreciate for Editors/Reviewers' warm work earnestly, and hope that the correction will meet with approval. Once again, thank you very much for your comments and suggestions. Look forward to hearing from you. 

Yours sincerely, 

Na WANG, Tele: 15827370668; E-mail: wangna@zgwhfe.com

May 29 2024,

Department of Breast，Wuhan Children’s Hospital（Wuhan Maternal and Child Healthcare Hospital），Tongji Medical College, Huazhong University of Science&Technology.

---

## [Decision Letter · Decision Letter 1]

26 Jul 2024

Clinical study of ultrasound-guided minimally invasive catheterization combined with compound cortex phellodendri fluid in the treatment of lactational breast abscess

PONE-D-24-05063R1

Dear Dr. Wang,

We’re pleased to inform you that your manuscript has been judged scientifically suitable for publication and will be formally accepted for publication once it meets all outstanding technical requirements.

Kind regards,

Zhaoqing Du, Ph.D

Academic Editor

PLOS ONE

Additional Editor Comments (optional):

Reviewers' comments:

Reviewer's Responses to Questions

**Comments to the Author**

1. If the authors have adequately addressed your comments raised in a previous round of review and you feel that this manuscript is now acceptable for publication, you may indicate that here to bypass the “Comments to the Author” section, enter your conflict of interest statement in the “Confidential to Editor” section, and submit your "Accept" recommendation.

Reviewer #2: All comments have been addressed

2. Is the manuscript technically sound, and do the data support the conclusions?

Reviewer #2: Yes

3. Has the statistical analysis been performed appropriately and rigorously? 

Reviewer #2: Yes

4. Have the authors made all data underlying the findings in their manuscript fully available?

Reviewer #2: Yes

5. Is the manuscript presented in an intelligible fashion and written in standard English?

Reviewer #2: Yes

6. Review Comments to the Author

Reviewer #2: (No Response)

7. PLOS authors have the option to publish the peer review history of their article (what does this mean?). If published, this will include your full peer review and any attached files.

Reviewer #2: No

---

## [Editor Report · Acceptance letter]

31 Jul 2024

PONE-D-24-05063R1 

PLOS ONE

Dear Dr. wang, 

I'm pleased to inform you that your manuscript has been deemed suitable for publication in PLOS ONE. Congratulations! Your manuscript is now being handed over to our production team.

Kind regards, 

on behalf of

Dr. Zhaoqing Du 

Academic Editor

PLOS ONE